# Nasal High-Flow (NHF) Improves Ventilation in Patients with Interstitial Lung Disease (ILD)—A Physiological Study

**DOI:** 10.3390/jcm12185853

**Published:** 2023-09-08

**Authors:** Jens Bräunlich, Marcus Köhler, Hubert Wirtz

**Affiliations:** Department of Respiratory Medicine, University of Leipzig, 04103 Leipzig, Germany; marcus.koehler87@googlemail.com (M.K.); hubert.wirtz@medizin.uni-leipzig.de (H.W.)

**Keywords:** nasal high-flow, NHF, hypercapnia, lung fibrosis, ILD, breathing pattern

## Abstract

Introduction: Acute hypercapnic respiratory failure has a poor prognosis in patients with interstitial lung disease (ILD). Recent data demonstrated a positive effect of nasal high-flow (NHF) in patients with acute hypoxemic respiratory failure. Preliminary data also show benefits in several hypercapnic chronic lung diseases. Objectives: The aim of this study was to characterize flow-dependent changes in mean airway pressure, breathing volumes, and breathing frequency and decreases in PCO_2_. Methods: Mean airway pressure was measured in the nasopharyngeal space. To evaluate breathing volumes, a polysomnographic device was used (16 patients). All subjects received 20, 30, 40, and 50 L/min and—to illustrate the effects—nCPAP and nBiPAP. Capillary blood gas analyses were performed in 25 hypercapnic ILD subjects before and 5 h after the use of NHF. Additionally, comfort and dyspnea during the use of NHF were surveyed. Results: NHF resulted in a small flow-dependent increase in mean airway pressure. Tidal volume was unchanged and breathing rate decreased. The calculated minute volume decreased by 20 and 30 L/min NHF breathing. In spite of this fact, hypercapnia decreased at a flow rate of 24 L/min. Additionally, an improvement in dyspnea was observed. Conclusions: NHF leads to a reduction in paCO_2_. This is most likely achieved by a washout of the respiratory tract and a reduction in functional dead space. NHF enhances the effectiveness of breathing in ILD patients by the reduction in respiratory rate. In summary, NHF works as an effective ventilatory support device in hypercapnic ILD patients.

## 1. Introduction

Interstitial lung disease (ILD) is characterized by a decrease in lung volumes, reduced diffusion capacity and interstitial changes in chest X-rays and CTs. The current classification describes several categories of idiopathic interstitial pneumonias: diffuse parenchymateous lung diseases with a known reason, granulomateous lung diseases, and rare entities such as lymphangioleimyomatosis or pulmonary Langerhans cell histiocytosis. For several years now, there have been drugs available with considerable effects on the fibrosing process [1,2].

One of the most prominent symptoms in ILD patients is breathlessness with hypoxemia or combined hypoxemia and hypercapnia when respiratory muscles are no longer able to keep up compensatory work. The application of long-term oxygen appears to be at least a symptomatic relief of dyspnea. Mechanical ventilation in end-stage disease without the perspective of listing for lung transplant is not advised and is followed by high mortality rates [3,4]. Some studies with mechanically ventilated hypercapnic ILD patients demonstrated improvements in oxygenation without modifying CO_2_ retention [3,5]. Non-invasive ventilation (NIV), including continuous positive airway pressure (CPAP) and bilevel pressure support, has only been tried in a few studies with little effect on the prevention of intubation (0–26%) [3,4,6,7]. However, two Japanese retrospective studies demonstrated improved survival when NIV (Bilevel or CPAP) was the first therapeutic option in acute exacerbated/progressive ILD [8,9]. A retrospective case series in eleven patients was published with a decrease in hypercapnia and improved oxygenation following the application of (bilevel) NIV [10]. Despite these studies, NIV is currently not recommended due to the absence of sufficient evidence of benefit in ILD patients, which is in strong contrast to patients with chronic obstructive pulmonary disease (COPD).

Nasal high-flow (NHF) is a new device providing warmed and humidified airflow supplemented with oxygen in case of need. Flow rates range up to 60 L/minute. Reduced intubation rates were found in the study by Frat et al. in acute hypoxemic respiratory failure mostly caused by pneumonia [11]. NHF showed non-inferiority in comparison to NIV in patients with a high risk of postextubation respiratory failure [12]. In addition, NHF has demonstrated superiority over conventional low-flow oxygen in a number of settings [13,14]. An important feature of NHF is the improved ability to maintain a high fraction of inspired oxygen (FIO_2_) in rapid breathing due to sufficient flow (20–60 L/min), thereby reducing most of the ambient air entrainment [15,16]. NHF changes breathing pattern and has led to improvements in blood gases in patients with acute or chronic respiratory failure [17,18,19,20]. NHF appears to be more effective in terms of pCO_2_ reduction with greater leak and higher flow rates [21]. Together, these beneficial effects, the increased elimination of pCO_2_ and the effective provision of large flows, if necessary, result in a decrease in the work of breathing, which is well suited to help patients with ILD and respiratory stress [18,19,20,22,23]. We therefore conducted this pilot study to evaluate nasal high-flow therapy in patients with ILD.

## 2. Material and Methods

### 2.1. Devices

In this study, we used the TNI softflow 50 device (TNI medical AG, Wuerzburg, Germany). NHF was applied using nasal prongs with different bore outlets (TNI medical AG, Wuerzburg, Germany) in order to apply the intended elevated flow rates. Small (ID 3.2 mm), medium (3.9 mm), and large (ID 5 mm) sizes of nasal prongs were compared as part of the study. nBiPAP and nCPAP were both applied using a nose mask (BiPAP Synchrony, Philips Respironics, Andover, MN, USA).

### 2.2. Subjects

A total of 25 ILD subjects in advanced stages of disease with hypercapnia were recruited from the respiratory ward at the university hospital of Leipzig from August 2015 to November 2017. All patients underwent long-term oxygen therapy. Of these subjects, 16 participated in the physiological tests as part of the protocol. The study was approved by the local ethics committee (414-14-06102014) and registered (ClinicalTrails NCT02504814). Subjects provided written informed consent to the trial protocol including the analysis of their data. The main exclusion criterion was an acute respiratory insufficiency.

The aim of this study was to gain a first impression on the effects of NHF versus conventional NIV in subjects with ILD and to characterize the influence of flow on ventilation and pCO_2_. Outcome parameters included changes in mean airway pressure, tidal volume (TV), and respiratory rate (RR). The clinical end-point was the change in paCO_2_ from baseline after 5 h of NHF breathing.

### 2.3. Measurement of Airway Pressure

A water-filled flexible tube (inner diameter 1 mm, Original Perfusor^®^-cable type standard, B. Braun, Melsungen, Germany) was placed in the nasopharyngeal space and then used as pressure transducer. A pressure sensor (GMH3111, Greisinger electronic GmbH, Regenstauf, Germany) connected to a laptop was used to record the signal. Ten breaths each were recorded during spontaneous, NHF, and nCPAP/nBiPAP breathing. Pressure during spontaneous breathing defined the baseline pressure (zero). Measurements were performed and repeated with different prong sizes. The measurements were carried out in the order shown. Data collection only took place after the baseline was reached again.

### 2.4. Measurement of Tidal Volume, Respiratory Rate, and Minute Volume

Sensor belts and a polysomnograph (Respitrace, Care Fusion GmbH, Höchberg, Germany) were used to measure tidal volumes. Subjects were measured in a sitting position. Elastic sensor belts were placed 10 cm below the jugular notch and 10 cm below the xiphoid process, and tidal volume measurements were performed. Subsequently, we calibrated the device individually for each subject, starting with “normal” tidal breathing which was recorded with standard lung function equipment (Master Screen Body, Care Fusion GmbH, Höchberg, Germany). While measuring tidal volumes of 10 breaths and simultaneously registering the sensor signal, we were able to calibrate the sensor belt signal to changes in lung volume. Following calibration, volume measurements during NHF and spontaneous breathing were started. Chest and abdominal excursions were recorded, and the medians in percental changes were used to calculate the volume in ml. Also, the minute volumes (MVs) were calculated. Measurements were performed and repeated with different prong sizes (small, medium, or large). The measurements were carried out in the order shown. Data collection only took place after the baseline was reached again.

### 2.5. Comfort and Dyspnea Scale

Subject impressions were correlated with various prong sizes and the use of nCPAP and nBiPAP. We used simple own dyspnea and comfort scales to evaluate patient satisfaction. We did not try to differentiate effects from flow rates, etc., on these scales. The comfort scale ranged from 1 to 10 and the dyspnea scale from 1 to 5. A smaller number defined more comfortable ventilatory support or less dyspnea.

### 2.6. Measurements in Hypercapnic ILD Subjects

Capillary blood gases were taken from the earlobe ten minutes after applying a hyperemia-inducing ointment (Finalgon^®^, Boehringer Ingelheim Pharma GmbH & Co. KG, Ingelheim am Rhein, Germany). Measurements were performed with constant oxygen flow before and after five hours NHF use. The flow rate was chosen for maximal tolerability.

### 2.7. Statistics

Data were analyzed using ANOVA (Sigma Plot, Systat Software GmbH, version 7, Ekrath, Germany). A probability level for the null hypothesis (no difference) of 5% or below (*p* < 0.05) was accepted for significance. The results were expressed as mean values ± SD. Changes compared to baseline values were also expressed as percentages because of the variation in the baseline values. Measurements following ventilatory support were compared with baseline values. Significance was defined as a significant change in percentage.

## 3. Results

Demographic data are presented below (Table 1). The group of subjects that underwent physiological measurements first exhibited a somewhat lower FVC compared to the entire group of subjects. All other parameters were not different. Not all subjects in the overall group agreed to or were able to undergo physiological measurements. All subjects performed the CO_2_ measurements.

### 3.1. Changes in Mean Airway Pressure

Mean airway pressure increased significantly with all settings and devices. During NHF breathing, the mean airway pressure increased in a flow-dependent manner. No differences were observed for different prong sizes. A flow of 20 L/min resulted in a significant but small increase in mean airway pressure (0.51 ± 0.33 mbar; *p =* 0.00). When the flow rate was increased to 50 L/min, the mean airway pressure increased to 3.1 ± 1.5 mbar; *p* = 0.00. nCPAP/nBiPAP led to an increase in mean airway pressure strongly dependent on the set pressure (nCPAP 6 mbar 5.0 ± 0.8 mbar; *p* = 0.00/nBiPAP 7.7 ± 1.3 mbar; *p* = 0.00) (Figure 1).

### 3.2. Changes in Breathing Patterns

TV did not change significantly during NHF breathing, while nCPAP/nBiPAP breathing went along with an increase in tidal volume (Figure 2). However, the respiratory rate decreased with NHF breathing, while during NIV breathing, the respiratory rate was unaltered (Figure 3). The minute volume was also significantly decreased with NHF at a flow rate of 20 and 30 L/min (Figure 4). Again, this was not observed with NIV. In contrast, a significant decrease in MV was observed during NIV breathing (Figure 4). We found no changes in the I/E ratio in all settings and devices.

### 3.3. Comfort and Dyspnea

Subjects felt more comfortable during the use of small NHF prongs independent of flow rates and rated dyspnea to be lower with small and medium NHF prongs. NIV breathing was associated with lesser comfort and more dyspnea compared to NHF (Table 2).

### 3.4. Changes in Partial Pressures of O_2_ and CO_2_

Treatment time in hypercapnic subjects was planned to be 5 h and was 4.7 ± 2.2 h in reality. Variation was organizational and treatment time had to be adapted to some extent to clinical schedules. The maximal tolerated flow rate was 24.0 ± 4.4 L/min. Oxygen was continuously delivered with a mean flow of 3.2 ± 1.5 L/min. PaO_2_ and pH were not different at the end of the NHF treatment periods in comparison with the baseline values, while paCO_2_ had decreased significantly (Table 3, Figure 5). 

## 4. Discussion

This is the first study evaluating and comparing the changes in breathing patterns and blood gases in subjects with ILD with ventilatory support from NHF and NIV. With NHF, a small increase in airway pressure was observed, as well as a reduction in respiratory rate and minute volume, and most importantly, a reduction in hypercapnia.

There is accumulating evidence on changes in breathing patterns in various situations of acute and chronic respiratory diseases during NHF breathing. It is mostly recognized that NHF leads to a decrease in respiratory rate and in minute volume [14,17,18,19,20,23,24]. The reduction in minute volume with no concomitant fall in pCO_2_ elimination is equivalent to a reduction in the work of breathing [18,20,22]. In contrast, tidal volumes vary in various conditions [17,19]. In patients with ILD, there was no change in tidal volume. This may be the consequence of the decreased compliance of the fibrotic lung. These findings are also in accordance with those in a previous study [17]. In contrast, NIV led to a significant increase in the tidal volumes, which surely is a consequence of alveolar recruitment and a reduction in the work of breathing. 

Similar to other studies, NHF led to a small increase in the mean airway pressure in our subjects [18,25]. This increase in airway pressure depended on the extent of leakage and on flow rates [21,25]. In our study, the increase in flow rates, but not prong size, influenced the mean pressure levels. In contrast to the differences between opening the mouth and keeping it closed, the change in prong size apparently did not alter leakage sufficiently to detect significant pressure changes. Some authors suggested alveolar recruitment as an important mechanism of action of NHF therapy [19]. The comparison of pressure levels with NHF and NIV ventilation and the resulting reductions in pCO_2_ in this study argue against airway pressure as an important mechanism at least in NHF ventilation in ILD patients. 

A reduction in pCO2 in various conditions with NHF breathing has been described in several studies [17,18,21,23,26]. However, most of these studies recruited a mixed normocapnic and hypercapnic population (pCO_2_ > 45 mmHg), and thus, the full potential of NHF to reduce hypercapnia cannot be recognized from these studies [11,14]. When exclusively hypercapnic patients were included, the pCO_2_ decrease during NHF breathing was more effective, and the extent of pCO_2_ reduction correlated to baseline PCO_2_ values [17,18,21,23,26]. This relation is best demonstrated for patients with COPD [18]. 

Increased patient comfort and a better tolerance might be suggested by our patient questionnaires. In chronic hypercapnic COPD, for example, NIV intolerance may be as high as 30% [27]. NHF is well tolerated, as reported in a number of studies, and, in general, results in increased use time compared to NIV [11,14,18,28]. The initiation of NHF support in ILD patients, in our experience, is significantly less time consuming than other methods of ventilatory support. However, there are no studies directly comparing instruction time requirements. 

NIV is not standard therapy in patients with interstitial lung disease but has been used in acute respiratory failure and may be helpful in a subset of patients without increased right ventricular load. However, the outcome of patients with IPF and acute respiratory failure (ARF) is poor, and more so with mechanical ventilation [3]. Our data show preliminary benefits with NHF support. NHF breathing should therefore be evaluated in a larger NHF cohort in direct comparison to NIV. 

### Limitations

One of the main limitations of the study is the lack of randomization and the fact that a number of patients did not tolerate the physiological tests. In addition, only paO_2_, but not FIO_2_ and SpO_2_, were documented during the hypercapnia reduction test. Furthermore, the authors point out that NIV is probably not the gold standard for comparisons in these patients with a risk of high expiratory tidal volume and ventilator-induced lung injury (VILI). Another limitation is the exploratory nature of this pilot study. Subject comfort and dyspnea scales were easy-to-understand school mark scales that were not validated prior to use. Thorax expansion was measured externally and translated into tidal volume. Due to individual compliance and the elastance of the thorax in each ILD subject, changes may have been over- or underestimated. The duration of NHF use varied to some extent, influenced mainly by the clinical schedule of ILD subjects. 

## 5. Conclusions

In summary, NHF breathing increased the mean airway pressure and changed breathing patterns in subjects with ILD. This led to a decreased respiratory rate but also a decrease in paCO_2_, demonstrating increased breathing efficiency. NHF might be a useful non-invasive ventilatory support in severe ILD patients with increased levels of paCO_2_.

## Figures and Tables

**Figure 1 jcm-12-05853-f001:**
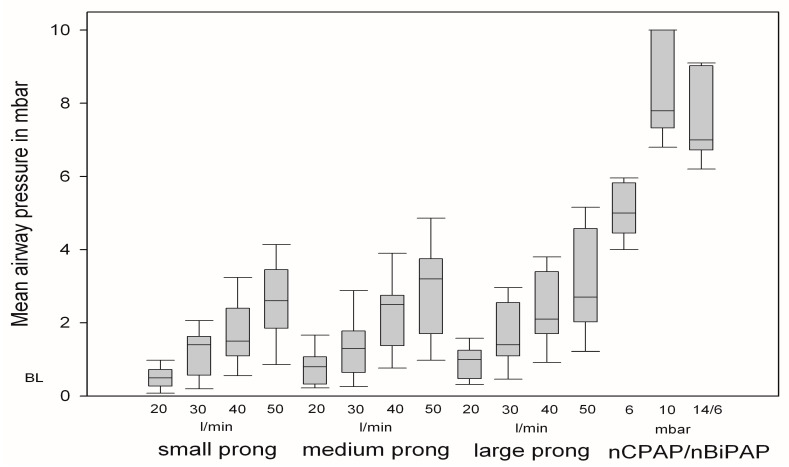
Mean airway pressure with different prong sizes and flow rates in comparison to non-invasive ventilation devices.

**Figure 2 jcm-12-05853-f002:**
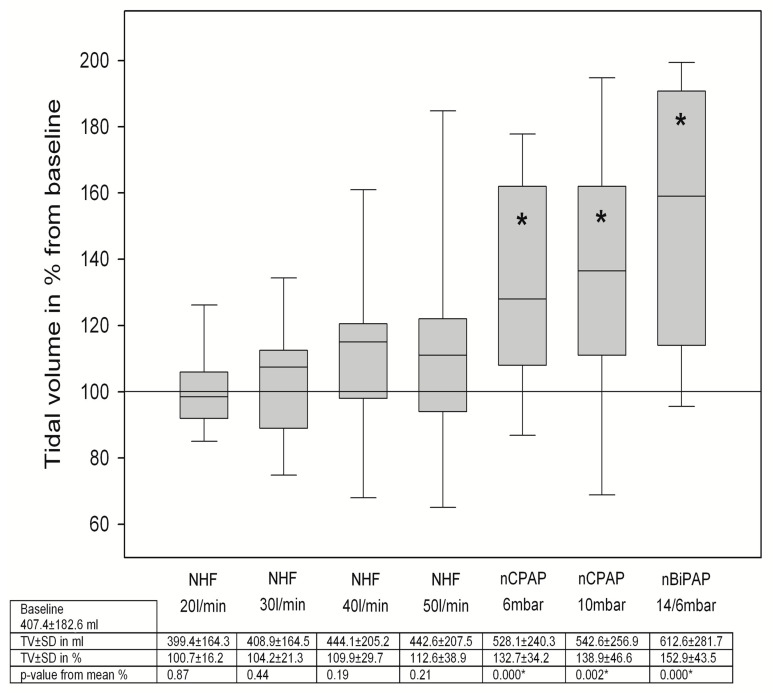
Changes in tidal volume during NHF and NIV breathing, TV = tidal volume, SD = standard deviation, * means *p* < 0.05.

**Figure 3 jcm-12-05853-f003:**
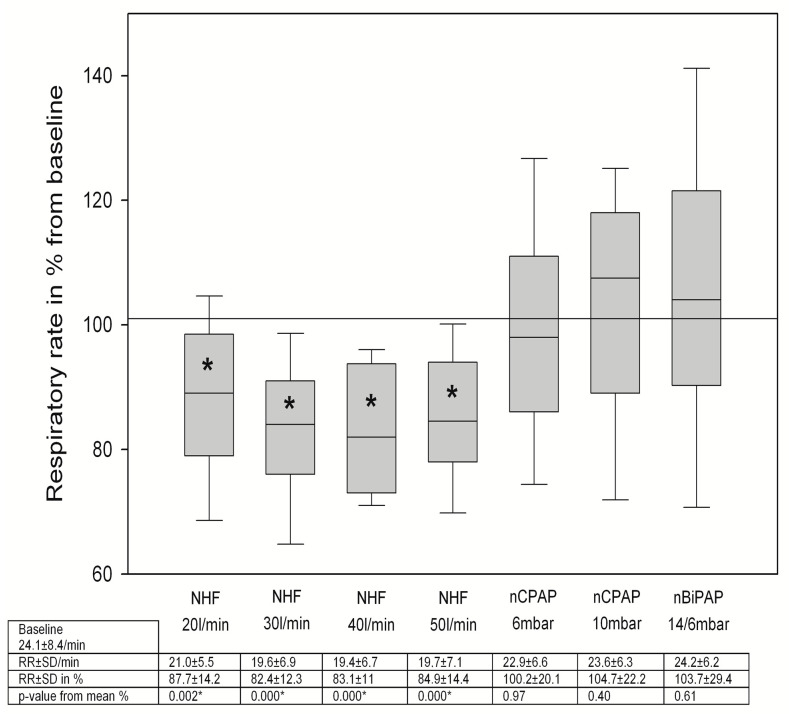
Changes in respiratory rate during NHF and NIV breathing, RR = respiratory rate, SD = standard deviation, * means *p* < 0.05.

**Figure 4 jcm-12-05853-f004:**
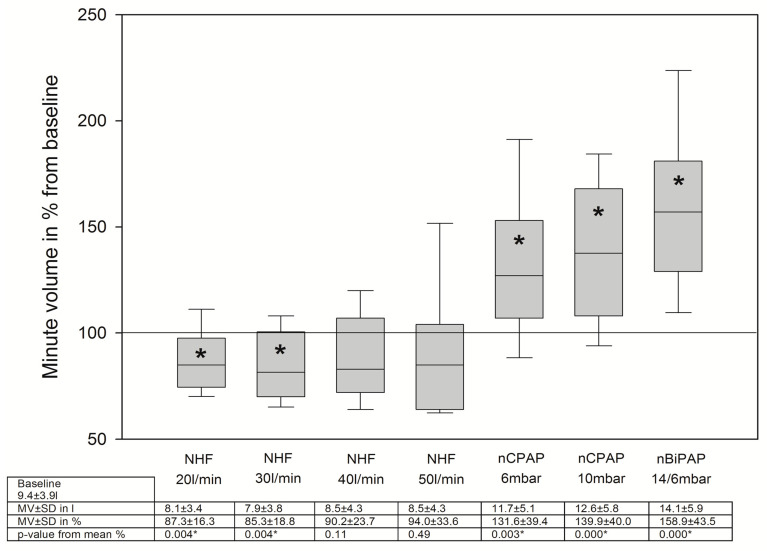
Changes in minute volume during NHF and NIV breathing, MV = minute volume, SD = standard deviation, * means *p* < 0.05.

**Figure 5 jcm-12-05853-f005:**
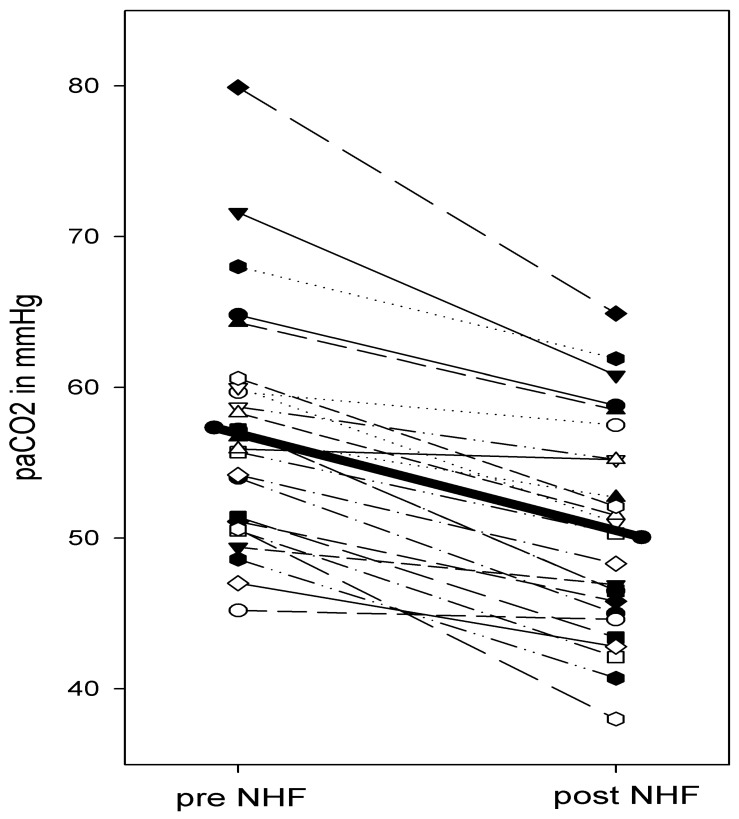
Changes in paCO_2_ during NHF breathing. Each thin line corresponds to one patient. The thick line shows the median value.

**Table 1 jcm-12-05853-t001:** Demographic data of physiological and hypercapnia group; * = *p* < 0.05; IPF = interstitial pulmonary fibrosis, EAA = extrinsic allergic alveolitis, LAM = lymphangioleiomyomatosis, PSS = primary Sjogren’s syndrome, FVC = forced vital capacity, FEV1/FVC = Tiffeneau index, TLC = total lung capacity.

	Physiological Group (*n* = 16)	All Patients (*n* = 25)
Female	10	18
Age (years)	59.1 ± 11.2	57.4 ± 8.3
Height (cm)	164.6 ± 8.2	166.3 ± 8.6
Weight (kg)	77.1 ± 17.7	80.3 ± 90.1
diagnosis	17 IPF, 4 sarcoidosis, 2 LAM, 1 PSS, 1 EAA	
Respiratory rate n/min	24.1 ± 8.4	24.8 ± 6.5
FVC % pred	50.0 ± 14.3 *	39.3 ± 10.8 *
FEV1/FVC %	90.1 ± 11.0	97.6 ± 19.4
TLC % pred	60.8 ± 11.9	56.2 ± 15.3

**Table 2 jcm-12-05853-t002:** Assessment of grade of comfort and dyspnea. Higher values means better comfort or lesser dyspnea, NHF = nasal high-flow, CPAP = continuous positive airway pressure, BiPAP = bilevel positive airway pressure.

	NHF Small Prong	NHF Medium Prong	NHF Large Prong	Nasal CPAP	Nasal BiPAP
Comfort	3.9 ± 1.8	4.0 ± 1.6	5.7 ± 2.1	5.9 ± 2.1	5.6 ± 2.5
Dyspnoe	2.9 ± 0.8	2.9 ± 0.8	3.1 ± 0.7	3.2 ± 0.9	3.4 ± 1.1

**Table 3 jcm-12-05853-t003:** paCO_2_ = arterialized partial pressure of carbon dioxide, paO_2_ = arterialized partial pressure of oxygen, NHF = nasal high-flow.

	pH	paCO_2_ mmHg	paO_2_ mmHg
Pre NHF	7.402 ± 0.07	57.2 ± 8.0	59.8 ± 10.5
Post NHF	7.434 ± 0.06	50.4 ± 7.2	58.7 ± 17.7
*p*-values pre/post NHF	0.17	0.003	0.83

## Data Availability

Data used in the current study are available from the corresponding author upon reasonable request.

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
