# Peer review of "Nasal High-Flow (NHF) Improves Ventilation in Patients with Interstitial Lung Disease (ILD)—A Physiological Study"

_jcm, 2023, doi:10.3390/jcm12185853_

Round 1

Reviewer 1 Report

This is an interesting study on the potential benefit of Nasal High Flow devices in a small group (n=25) of patients with ILD. It is limited by the small numbers but is recognised as a pilot study. The study is well conducted and has CPAP and NIV as comparators to NHF. 16/25 of the patients undergo PSG for further physiological monitoring. All patients would be regarded as having severe ILD in context of being hypercapnic.

The authors report a reduced Respiratory rate, but reduced PCO2 and no change in tidal volume with use of NHF. This is believed to be due to reduction in worrk of breathing and  improved CO2 washout by the higher flows.

The reduction in PCO2 is recorded for NHF but not for CPAP or BiPAP. This would put the use of NHF as a ventilatory support device into context. It is well reported that NHF is well tolerated but is it as effective as NIV for example at improving ventilation. 

The study is not controlled and it would be useful to have had a concurrent matched control group such as COPD where its use is perhaps a little more common.

The trial of each flow rate were for 5 hours and It would be worthwhile explaining how this duration was chosen over shorter or longer time periods.

I feel there needs to be a review of the English and I have made a few comments about this in the section below.

Otherwise it is an interesting study which would need further larger scale investigation and ideally a control disease group.

Line 34  change perspective to prospect of or potential for

Line 44 delete of cause or replace with of course. 

Line 29 I suggest rewording the sentence on drugs in ILD as considerable effects is very vague and its unclear as to whether referring to clinical improvements or adverse effects.

Line 55 change was led to has led

Line 56 consider changing leakage to leak

Line 126 not all patients were agreed - suuggest change to not all patients agreed to or were able to..

Line 148 change decreased significant to a significant decrease was observed

Line 190 change increased insignificantly to there was no change in tidal volume. Insignificantly could be misread or misinterpreted for signifcantly.

Line 191 I would suuggest also a comment on how CPAP results in increase in tidal volume and the likely mechanism. (is this due a degree of alveolar recruitment and reduction in work of breathing).

Author Response

Many thanks for the helpful comments, which have all been implemented.

The study was designed as a physiological study and not as a comparison of the effectiveness of CO2 reduction of the different devices. Therefore, there is no comparison between the different devices. Regarding the effects in a control group (COPD), a paper with the same experimental design exists (PMID 27307723). The duration of NHF use in hypercapnic patients was calculated as a possibly sufficient period. Unfortunately, there are no data on the time period in which a sufficient decrease in CO2 can be expected.

Reviewer 2 Report

I am thankful for giving me the chance to review the manuscript entitled “Nasal High-Flow (NHF) Improves Ventilation in Patients with Interstitial Lung Disease (ILD)”. The topic seemed with little novelty. The main idea/content is to compare the effects on ventilation between NHF and BiPAP, that is good. However, the most important data of the change of PaCO2 between NHF and BiPAP is lacking. The authors could just report the difference of mean airway pressure, tidal volume, respiratory rate, minute ventilation, and grade of comfort and dyspnea between NHF and BiPAP, making the study nothing new. 

1.    Title: The authors could mention which type of study it is.

2.    Introduction (2nd paragraph): NIV is not a specific device. The authors may describe in detail about what kind of NIV in reference #3, 4, 6, 7, 8, 9 and 10.

3.    Subjects: Do the authors mean 16 patients received complete protocol; other 9 patients received only PaCO2 analysis? Why? Maybe, the authors should report the protocol in the section.

4.    Subjects: “The aim of this study was to get a first impression on the effects of NHF versus conventional BiPAP in subjects with ILD and to characterize the influence of flow on ventilation and PaCO2.” Do the authors mean these tests (NHF, CPAP, BiPAP) were performed in the same subject during different time? What is the interval? What is the sequence (all NHF, CPAP, then BiPAP)? How do the authors make sure that the subject return to baseline breathing without the effects of previous test?

5.    Following the aim of the study, the most important test is to compare the change of PaCO2 in subjects using NHF versus BiPAP. However, the authors only present data of before and after NHF. This made the data uselessly. As we know, mean airway pressure, tidal volume, respiratory rate, minute ventilation, and even grade of comfort and dyspnea should be different between NHF and BiPAP since they have different designs and mechanisms.

6.    The role of CPAP was not well mentioned in the study.

7.    Why do the authors report with “nCPAP/nBiPAP” in figure 1? The authors should report nCPAP (6 and 10) and nBiPAP (14/6) such as other figures.

Author Response

Many thanks for the helpful comments.

  1. The title was adapted.
  2. A clarification was introduced (line 38).
  3. Line 130 ( contains the following sentence: “Not all subjects in the overall group agreed to or were able to undergo physiological measurements.” I added the sentence “All subjects performed the CO2 measurements.”
  4. Many thanks, of course it is important to clarify this point. In both sections I added the sentence: “The measurements were carried out in the order shown. Dat collection only took place after the baseline was reached again.”
  5. The aim of the study was not to compare BiPAP and NHF. The NHF was shown to be physiologically effective in ILD patients. It is important information and not generally accepted that NHF can lower CO2. Therefore, I do not see the data as uselessy.
  6. The role of CPAP was included in the interpretation of NIV.
  7. Due to the large amount of data in this graph, a more condensed presentation was necessary. The focus is on the prong sizes used. The representation 6mbar,10mbar and 14/6 mbar was retained.

Reviewer 3 Report

The manuscript submitted for review focuses on the effects of high-flow nasal therapy in patients with interstitial lung disease. The data provided by the study are interesting, but there are certain points that require clarification.

MAJOR ISSUES

- Type of patients included in the study: The phenomenon of hypercapnia is uncommon in patients with interstitial disease, and usually occurs in very advanced stages of the disease or when there is another reason for the hypercapnia.

- It is not reflected whether the level of hypercapnia was based on arterial blood gas or earlobe capillary blood gas.

- No data are provided on the number of patients under treatment with home oxygen therapy. In addition, although no patients are included later, the legend in Table 1 includes cystic fibrosis as an interstitial disease, which is not correct.

- Similarly, it is not clear at what FiO2 the test was performed with high-flow nasal therapy or ventilation.

- It is not clear why only 16 patients were selected for the physiological study. It is implied in the results that the rest of the patients did not give consent. This should be reflected in the material and methods.

- In the material and methods section it is stated that pressure was continuously monitored at the pharyngeal level. Mean pressure data are provided, but it would also be interesting to know what the minimum pressure values were, to determine whether there was a decrease in pressure swings associated with inspiratory effort under nasal therapy or noninvasive ventilation.

- Inductive plethysmography data are provided but it is not specified how the device was calibrated (we understand that it was calibrated against a spirometer). What was the value considered as baseline (the sum of both belts?). It would also be interesting to know if there were changes in the ventilatory pattern measured by bands (ratio inspiratory time / total time, individual components of the bands (RC/AB components vs sum, etc). Finally, how was the sum of bands calculated in the presence of thoracoabdominal incoordination, if any?

MINOR POINTS.

- Line 110: the number of patients should be included in results, not in material and methods.

Author Response

Many thanks for the helpful comments.

  1. There were no other reasons for hypercapnia. The patients were in a very advanced stage and on the transplant list. This explains the hypercapnia. I added the note “in advanced stage of disease” in line 74.
  2. Measurement of blood gases was explained in detail in line 119 as follows: “Capillary blood gases were taken from the earlobe ten minutes after applying a hyperemia-inducing ointment (Finalgon®, Boehringer Ingelheim Pharma GmbH & Co. KG, Ingelheim am Rhein, Germany).”
  3. Very important. Thank you! As you can see in table 1 no CF patient was included. It was a mistake in the legend. We adapted the legend. All patients had LTOT. I made a note on line 76.
  4. The FIO2 was not documented. We wrote in line 122: “Measurements were performed with constant oxygen flow before and after five hours NHF use.” As seen in table 2 pO2 was constant before and after the 5 hours period. This was the aim.
  5. We would prefer to mentioned this fact in the results section as follows: line 135 contains now: “Not all subjects in the overall group agreed to or were able to undergo physiological measurements. All subjects performed the CO2 measurements.“
  6. We have not documented these parameters.
  7. We measured in percent of changes
  8.  

Value at baseline: We measured excursions at quite breathing (same as values in spirometry 0 baseline volume in ml). Then we measured changes in percent and could calculate the volume in ml. Clarification in line 108 as follows: “Chest and abdominal excursions were recorded and the median in percental changes were used to calculate the volume in ml.“

Ratio: Line 161: We found no changes in I:E ratio in all settings and devices.

Other patterns were not recorded. And inccoordinations were not present despite of an end-stage disease.

Minor: We removed the sentence.

Round 2

Reviewer 2 Report

The authors did not answer most of the reviewer's questions.  The details are as follows:

2: Introduction (2nd paragraph): NIV is not a specific device. The authors may describe in detail about what kind of NIV in reference #3, 4, 6, 7, 8, 9 and 10.
2': The authors still do not show which study uses which device.

3: Subjects: Do the authors mean 16 patients received complete protocol; other 9 patients received only PaCO2 analysis? Why? Maybe, the authors should report the protocol in the section.
3': The authors do not answer the main question: why did the 9 patients not receive complete protocols? In my opinion, they should be excluded if they could not follow the total protocols. Furthermore, their data may influence the overall data in CO2 measurements. In addition, the authors should describe details in methods, not in results.

5: Following the aim of the study, the most important test is to compare the change of PaCO2 in subjects using NHF versus BiPAP. However, the authors only present data of before and after NHF. This made the data useless. As we know, mean airway pressure, tidal volume, respiratory rate, minute ventilation, and even grade of comfort and dyspnea should be different between NHF and BiPAP since they have different designs and mechanisms.
5': The authors said that "The aim of this study was to get a first impression on the effects of NHF versus conventional BiPAP in subjects with ILD and to characterize the influence of flow on ventilation and pCO2. (Lines 79-80)". In my opinion, the test to compare the change of PaCO2 in ILD subjects using NHF versus BiPAP is critical. The measurement, by using the change in PaCO2 between baseline and after 5 hours of NHF breathing, did not match the aim of the study. The authors should modify their aim of the study if they only performed the present measurements.

6: The role of CPAP was not well mentioned in the study.
6': The authors said that "The aim of this study was to get a first impression on the effects of NHF versus conventional BiPAP in subjects with ILD and to characterize the influence of flow on ventilation and pCO2. (Lines 79-80)". CPAP was not included in the aim. This is not correct.

7: Why do the authors report with “nCPAP/nBiPAP” in figure 1? The authors should report nCPAP (6 and 10) and nBiPAP (14/6) such as other figures.
7': It is obvious that there is enough space to delete the "/" and use a "space" to divide nCPAP and nBiPAP.

Author Response

Dear reviewer,   thank you for the helpful comments.   Q2: The studies cited are usually observational studies that looked at different forms of ventilation in ILD. Here, both bilevel and CPAP were used in the studies and considered together (described for 3,4,6,7). This was now specified again for reference 8,9,10. A more detailed specification would not provide much better information for the reader and would rather distract from the topic.   Q3: As described in line 137, the patients were not able or did not want to do the examinations. This is quite understandable due to the severity of the disease. Please note that this is not an RCT in which the patients would be excluded, but represents a physiological study. In our opinion, this statement belongs in the results section, as this was not planned.   Q5/6: "The aim of this study was to get a first impression on the effects of NHF versus conventional BiPAP in subjects with ILD and to characterize the influence of flow on ventilation and pCO2.". This description describes exactly what was studied. Ventilatory effects were measured in NIV and NHF and the influence of flow on CO2 was investigated. We changed BiPAP to NIV.   Q7: Figure 1 was adapted

Reviewer 3 Report

The authors adequately addressed my previous concerns. 

Author Response

Thank you